# Thermoelectric Cycle and the Second Law of Thermodynamics

**DOI:** 10.3390/e25010155

**Published:** 2023-01-12

**Authors:** Ti-Wei Xue, Zeng-Yuan Guo

**Affiliations:** Key Laboratory for Thermal Science and Power Engineering of Ministry of Education, Department of Engineering Mechanics, Tsinghua University, Beijing 100084, China

**Keywords:** the second law of thermodynamics, theorem of the equivalence of transformations, reversible thermoelectric cycle, combined power–refrigeration cycle

## Abstract

In 2019, Schilling et al. claimed that they achieved the supercooling of a body without external intervention in their thermoelectric experiments, thus arguing that the second law of thermodynamics was bent. Kostic suggested that their claim lacked full comprehension of the second law of thermodynamics. A review of history shows that what Clausius referred to as the second law of thermodynamics is the theorem of the equivalence of transformations (unfairly ignored historically) in a reversible heat–work cycle, rather than “*heat can never pass from a cold to a hot body without some other change*” that was only viewed by Clausius as a natural phenomenon. Here, we propose the theorem of the equivalence of transformations for reversible thermoelectric cycles. The analysis shows that the supercooling phenomenon Schilling et al. observed is achieved by a reversible combined power–refrigeration cycle. According to the theorem of equivalence of transformations in reversible thermoelectric cycles, the reduction in body temperature to below the ambient temperature requires the body itself to have a higher initial temperature than ambient as compensation. Not only does the supercooling phenomenon not bend the second law, but it provides strong evidence of the second law.

## 1. Introduction

In 2019, Schilling et al. [1], from the University of Zurich, performed an interesting thermoelectric experiment. They used a “thermal inductor”, composed of a Peltier element and electric inductance, to cool a body with an initial temperature above ambient all the way down to below ambient temperature, as shown in Figure 1. Schilling et al. then concluded that “*the use of a thermal connection containing a variant of a thermal inductor can drive the flow of heat from a cold to a hot body without external intervention”*. This claim referenced the words of the famous Clausius statement (*heat can never pass from a cold to a hot body without some other change*) of the second law of thermodynamics, but gave what appeared to be the exact opposite view. They explained that “*the experiments appear to be a kind of thermodynamic magic, thereby challenging to some extent our traditional perceptions of the flow of heat”* [2]. Nevertheless, Schilling et al. [3] emphasized that the experiments did not break the second law of thermodynamics, but only bent it. They additionally calculated entropy production during the experiments to support their opinion.

Schilling et al.’s experiments are real, ingenious and can provide inspiration for engineering practice. Yet, their interpretations were somewhat tendentious, mysterious and did not clearly reveal the thermodynamic principles implied in the experiments. According to Kostic [4], this “*is mostly due to the lack of full comprehension of the still elusive second law of thermodynamics and subtle issues related to thermal phenomena*”. Kostic critically analyzed and interpreted Schilling et al.’s unreasonable claims in detail and scientifically. Kostic pointed out that the claim, “*heat flow from cold to hot without external intervention*”, was false. He considered that the “external intervention” in the experiments is simply the EM energy stored in the inductor (external to the Peltier element). Kostic further noted that “*work could have been achieved by any work generating device, stored by any suitable device, and such stored work used subsequently in any refrigeration device to sub-cool the body”*.

It seems that a full understanding of the connotation of the second law of thermodynamics is critical to the interpretation of Schilling et al.’s experimental results. As Schilling himself said, the cognition “*may have been misguided by certain shortened versions of the second law of thermodynamics in some textbooks*” [3]. The Clausius statement of the second law of thermodynamics is generally considered to be “*heat can never pass from a cold to a hot body without some other change*” in most current textbooks. However, a review of history revealed that Clausius viewed it only as a natural phenomenon [5]. Similar phenomena can be found in other disciplines. For example, an object can never move from a low to high location in a gravity field without some other change, and electrical charges can never move from a low to high potential in an electrostatic field without some other change. What Clausius originally referred to as the second law of thermodynamics was in fact the theorem of the equivalence of transformations in a reversible thermodynamic cycle [5,6,7,8]. Analyses show that the thermodynamic principles involved in Schilling et al.’s thermoelectric experiments could be well clarified by the idea of reversible cycle and the theorem of equivalence of transformations. The supercooling phenomenon in Schilling et al.’s thermoelectric experiments was not achieved by a single thermodynamic cycle, but by a reversible combined power–refrigeration cycle. According to the theorem of the equivalence of transformations, the negative-going flow of heat from the colder body to the ambient in the refrigeration cycle must be compensated by the positive-going flow of heat from the body with a higher initial temperature to the ambient in the power cycle.

## 2. Clausius’ Original Version of the Second Law of Thermodynamics

Clausius’ work on the second law of thermodynamics was originally intended to improve Carnot’s theorem [7,8]. In 1824, Carnot [9] famously stated that there is a certain relationship independent of the nature of the working fluid between heat transfer and work production in a steam engine (later called Carnot’s theorem). However, Carnot was unable to determine this relationship due to a misperception of heat at that time. By 1850, the efforts of Rumford, Mayer, Joule and Helmholtz had finally succeeded in eliminating this misconception [10]. Clausius summarized it as the theorem of equivalence of heat and work, also known as the first law of thermodynamics. Four years later, Clausius succeeded in reconciling Carnot’s theorem with the first law of thermodynamics. He pointed out that there were two kinds of transformations, i.e., the transformation of heat to work and the transformation of heat from a high temperature to a low temperature in a reversible power cycle [7,8,11]. Of the two transformations in such a reversible process, either can replace the other if the latter is taken in an opposite direction; therefore, if a transformation of one kind has occurred, this can be again reversed, and a transformation of the other kind may be substituted without any other permanent change being requisite thereto. Therefore, these two kinds of transformations can be regarded as phenomena of the same nature and are thus equivalent, referred to as the theorem of the equivalence of transformations by Clausius [7,8]. According to Clausius, “*Carnot’s theorem actually expressed a relation between the two kinds of transformations”* [7]. The theorem of the equivalence of transformations is in-principle based on the Carnot’s “reversible equivalency” [12].

Clausius further assigned the “equivalence-values” to the two kinds of transformations by employing a thermodynamic cycle with three heat sources, as shown in Figure 2 [7,8]. The equivalence-value for the transformation of work to heat (Inverse of Process f~a) was expressed as *Qf* (*t*), where *f* (*t*) is a function of temperature, *t* (the concept of absolute temperature was not yet established at the time). The equivalence-value for the transformation of heat from a high temperature to a low temperature (Combination of Processes b~c and Processes d~e) was expressed as *Q*_2_*F*(*t*_1_,*t*_2_), where *F*(*t*_1_,*t*_2_) is a function of temperature *t*_1_ and *t*_2_. By means of a fundamental principle, i.e., *heat can never pass from a cold to a hot body without some other change*, Clausius then demonstrated that the sum of the equivalence-values for all transformations in the reversible thermodynamic cycle was zero [5,7,8], as follows:(1)$$-Qf\left(t\right)+Q_{2}F\left(t_{1},t_{2}\right)=0$$

Equation (1) is the mathematical expression of the theorem of the equivalence of transformations. Further analysis yielded a relationship between two functions of temperature [5,7,8]:(2)$$F\left(t_{1},t_{2}\right)=f\left(t_{2}\right)-f\left(t_{1}\right)$$

Thus, for a reversible thermodynamic cycle with a series of continuous heat sources, the sum, *N*, of the equivalence-values of all transformations can be expressed as:(3)$$N=\oint f\left(t\right)dQ=0$$

Equation (3) implies that the theorem of the equivalence of transformations holds as long as the cycle is reversible, regardless of its form. Clausius then derived that the integration factor, *f* (*t*), was exactly the reciprocal temperature, 1/*T* [13]. Thus, Equation (3) becomes:(4)$$\oint \frac{1}{T}dQ=0$$

Clausius called Equation (4) the mathematic expression of the second law of thermodynamics (later named Clausius equality) [7,8]. It is worth remarking that Equation (4) suggests the appearance of a state variable. In 1865, Clausius officially named this state variable as “entropy”, *S*:(5)$$dS=\frac{dQ_{\mathrm{rev}}}{T}$$
where the subscript, rev, denotes reversible. Based on the theorem of the equivalence of transformations, Clausius finally made clear the expression for Carnot efficiency, *η* [5]:(6)$$\eta =1-\frac{T_{2}}{T_{1}}$$
where *T*_1_ is the temperature of the heat reservoir and *T*_2_ is the temperature of the heat sink.

Clausius’ work brought symmetry and beauty to the axiomatic framework of thermodynamics [14]. Both from a linguistic and a conceptual point of view, Clausius intentionally paralleled the theorem of the equivalence of transformations with the theorem of equivalence of heat and work. These two theorems of equivalence express two aspects of the same thing. The theorem of equivalence of heat and work refers to the quantitative relationship between heat and work, corresponding to the conservation of internal energy, while the theorem of the equivalence of transformations refers to the quantitative relationship between the two kinds of transformations, corresponding to the conservation of entropy. The theorem of equivalence of heat and work is a manifestation of the first law of thermodynamics. The theorem of the equivalence of transformations was considered as the second law of thermodynamics by Clausius [7,8]. Note that “*heat can never pass from a cold to a hot body without some other change*” had its roots in experience alone and had no other support, except that it was consistent with the first law [15]. This statement, as a natural phenomenon, was only used by Clausius as a fundamental principle to prove the theorem of the equivalence of transformations [5]. It guarantees that the sum of the equivalence-values for all transformations in a reversible thermodynamic cycle must be zero. If the sum is not zero, then a net uncompensated negative transformation of heat from a low temperature to a high temperature can always be found, which contradicts the basic principle.

## 3. Theorem of the Equivalence of Transformations for a Thermoelectric Cycle

Clausius had already established the basic concepts and laws of equilibrium thermodynamics in his unique way. However, Clausius’ interests were confined to the theory of thermodynamic cycle for heat–work conversion [15]. The idea of cycles has not been discussed in such depth in thermoelectric conversion as in heat–work conversion, perhaps because the cycle forms for thermoelectric conversion are usually not easily identified or described. Nevertheless, some concepts and conclusions from the heat–work cycle have always been applied to thermoelectricity, despite being verified to be equally valid. For example, thermoelectric devices are also restricted by the Carnot efficiency [16,17,18]. Many thermodynamic conclusions, including Carnot efficiency, are natural outcomes of Clausius’ theorem of the equivalence of transformations for reversible heat–work cycle. Therefore, it is necessary to establish a theorem of the equivalence of transformations for a reversible thermoelectric cycle to fill the missing link in the logic chain.

In the way of Clausius, there should be two kinds of transformations in a reversible thermoelectric power cycle: (a) the transformation of heat from a high temperature to a low temperature; (b) the transformation of heat to electrical work. Of the two transformations in such a reversible process either can replace the other if the latter is taken in an opposite direction; so that if a transformation of the one kind has occurred, this can be again reversed, and a transformation of the other kind may be substituted without any other permanent change being requisite thereto. Therefore, these two kinds of transformations can be regarded as phenomena of the same nature and are thus equivalent. The transformation of electrical work to heat and correspondingly, the transformation of heat from a high temperature to a low temperature, are positive, and the opposites are negative. As with a reversible heat–work cycle, based on Clausius’ fundamental principle, it can be proven that the sum of the equivalence-values for all transformations in a reversible thermoelectric cycle is zero. Similarly, the theorem of the equivalence of transformations holds as long as the thermoelectric cycle is reversible, independent of its form. Based on the theorem of the equivalence of transformations for heat–work cycles, Clausius obtained many important conclusions of classical thermodynamics, including the expressions for the Carnot function and the Carnot efficiency, the Clausius equality and the definition of entropy. The same conclusions can be obtained from the theorem of the equivalence of transformations for thermoelectric cycles. Therefore, it can be considered that thermoelectric systems follow the same thermodynamic principles as heat–work systems.

In 1856, William Thomson (also known as Lord Kelvin) obtained the well-known Kelvin relation for thermoelectricity [19,20,21]. However, his research came under heavy criticism due to the lack of support from recognized fundamental theories [22,23]. The idea of reversible cycle and the theorem of equivalence of transformations, as the thermodynamic bases of thermoelectricity, might be able to provide a rebuttal to the criticism. Kelvin conceived of a thermoelectric circuit comprising two kinds of materials, A and B, as shown in Figure 3. There are two contact junctions between the two materials and they are in contact with heat sources of different temperatures [21]. Kelvin pointed out that when a unit charge travels around the circuit, the algebraic sum of all forms of energy discharged from the circuit per unit time should be equal to zero [20]:(7)$$Is\Delta T+I\pi \left(T\right)-I\pi \left(T+\Delta T\right)-I\int_{T}^{T+\Delta T}\left(\sigma_{B}-\sigma_{A}\right)dT=0$$
where *s* is the Seebeck coefficient, *π* is the Peltier coefficient, and *σ* is the Thomson coefficient. Kelvin further pointed out that if a material system experiences a complete cycle of a perfectly reversible kind, the quantities of heat which it takes in at different temperatures are subject to a linear equation, of which the coefficients is the reciprocal of the temperature [20]. For the reversible thermoelectric circuit, the contributions to heat from the Peltier effect and from the Thomson effect cancel each other out [24]:(8)$$I\frac{\pi \left(T\right)}{T}-I\frac{\pi \left(T+\Delta T\right)}{T+\Delta T}-I\int_{T}^{T+\Delta T}\frac{\sigma_{B}-\sigma_{A}}{T}dT=0$$

Combining Equations (7) and (8) yielded the Kelvin relation, as follows:(9)$$s=\frac{\pi }{T}$$

Kelvin drew on the ideas of Joule, Carnot, and Clausius, etc., to deal with the problem of thermoelectricity [19]. Kelvin’s thermoelectric circuit is essentially a reversible thermodynamic cycle. Equation (7) is a manifestation of the first law of thermodynamics in the thermoelectric cycle. Equation (8) is the mathematical expression of the theorem of the equivalence of transformations (the second law of thermodynamics as referred to by Clausius) in the thermoelectric cycle. It expresses that the net change of entropy for a reversible thermoelectric cycle is zero (note that when Kelvin derived Equation (8), the concept of entropy had not yet been established) [24,25]. In other words, Equation (8) is the equality in the thermoelectric cycle, similar to Clausius equality in the heat–work cycle. This explains how Kelvin gets the correct Kelvin relation.

## 4. Theorem of the Equivalence of Transformations for a Combined Power–Refrigeration Cycle

The general operation of a combined power–refrigeration cycle is as follows. The heat engine absorbs heat from a heat reservoir, converts some of it to (electrical) work, and outputs the remainder to a heat sink. All of the converted (electrical) work is then used to drive a heat pump/refrigerator to absorb heat from a low-temperature heat reservoir and deliver it to a high-temperature heat sink. The theorem of the equivalence of transformations holds for a reversible thermodynamic cycle. Since the cycle is reversible, the theorem of the equivalence of transformations holds for its inverse as well, and further holds for the combined cycle consisting of the cycle and its inverse. For a combined cycle, (electrical) work is generated in the heat engine and then completely consumed by the heat pump/refrigerator. That is, (electrical) work only acts as a medium. Therefore, a combined cycle can be fully expressed as a combination of several transformations of heat between different temperatures. In other words, a combined cycle achieves the transformation of the quantity and temperature (grade) of heat, which is similar to the principle of a transformer in electricity; therefore, a combined cycle can be considered as a “thermal transformer” [4,26]. As shown in Figure 4, the power cycle part in the combined cycle contains the transformation of *Q*_1_ from *T*_1_ to *T*_2_, *Q*_1_*F*(*T*_1_,*T*_2_), and the transformation of *Q*_m_ at *T*_1_ to electrical work, −*Q*_m_*f*(*T*_1_) and the refrigeration cycle part contains the transformation of *Q*_2_ from *T*_4_ to *T*_3_, *Q*_2_*F*(*T*_4_,*T*_3_) and the transformation of electrical work to *Q*_m_ at *T*_3_, *Q*_m_*f*(*T*_3_). According to Equation (2), *Q*_m_*F*(*T*_1_,*T*_3_) = *Q*_m_*f*(*T*_3_) − *Q*_m_*f*(*T*_1_). That is, the transformation of *Q*_m_ at *T*_1_ to electrical work in the power cycle part and the transformation of electrical work to *Q*_m_ at *T*_3_ can be combined into one transformation of *Q*_m_ from *T*_1_ to *T*_3_. Therefore, the reversible combined cycle can be regarded as containing three transformations, i.e., (1) the transformation of *Q*_1_ from *T*_1_ to *T*_2_; (2) the transformation of *Q*_m_ from *T*_1_ to *T*_3_; (3) the transformation of *Q*_2_ from *T*_4_ to *T*_3_. These three transformations cancel each other out; thus, the algebraic sum of the equivalent values of all transformations is zero:(10)$$Q_{1}F\left(T_{1},T_{2}\right)+Q_{m}F\left(T_{1},T_{3}\right)+Q_{2}F\left(T_{4},T_{3}\right)=0$$

This is the theorem of the equivalence of transformations for a reversible combined power–refrigeration cycle.

Schilling et al. cooled a body with an initial temperature above ambient all the way down to below ambient temperature in their thermoelectric experiments. The supercooling phenomenon cannot be achieved by a single thermodynamic cycle, but instead requires a reversible combined power–refrigeration cycle. There are two stages for Schilling et al.’s combined cycle, as shown in Figure 5. In the first stage, the thermoelectric circuit goes through a power cycle to absorb heat from the body with a finite heat capacity. In this stage, the body acts as the heat reservoir and the ambient acts as the heat sink. Electrical work is continuously generated and stored in the inductor until the body drops to ambient temperature during this stage. In the second stage, the thermoelectric circuit goes through a refrigeration cycle and the electrical work previously accumulated in the inductor re-enters and drives the thermoelectric circuit to continue to absorb heat from the body and release it to the ambient. In this stage, the body acts as the low-temperature heat reservoir and the ambient acts as the high-temperature heat sink. Thus, the temperature of the body is further reduced to below the ambient temperature until the electrical work is completely consumed.

Schilling et al.’s combined cycle contains three transformations of heat from a high temperature to a low temperature, i.e., (1) the transformation of *Q*_1_ from *T*_b,1_ to *T*_r_; (2) the transformation of *Q*_m_ from *T*_b,1_ to *T*_r_; (3) the transformation of *Q*_2_ from *T*_b,2_ to *T*_r_, as shown in Figure 6a. Thus, the theorem of the equivalence of transformations is expressed as:(11)$$Q_{1}F\left(T_{b,1},T_{r}\right)+Q_{m}F\left(T_{b,1},T_{r}\right)+Q_{2}F\left(T_{b,2},T_{r}\right)=0$$

Since the ambient acts as the heat sink for both the power cycle and the refrigeration cycle, the combined cycle can be simplified to contain only two transformations, i.e., (1) the transformation of (*Q*_1_ + *Q*_m_) from *T*_b,1_ to *T*_r_; and (2) the transformation of *Q*_2_ from *T*_b,2_ to *T*_r_, as shown in Figure 6b. Thus, the theorem of the equivalence of transformations is simplified as:(12)$$\left(Q_{1}+Q_{m}\right)F\left(T_{b,1},T_{r}\right)+Q_{2}F\left(T_{b,2},T_{r}\right)=0.$$

The transformation of (*Q*_1_ + *Q*_m_) from *T*_b,1_ to *T*_r_ is positive and the transformation of *Q*_2_ from *T*_b,2_ to *T*_r_ is negative. The occurrence of the negative transformation must be compensated by the positive transformation. In other words, heat could pass from the colder body to the ambient, but before that can happen, the body itself needs to have a higher temperature than the ambient as compensation.

Consider the following two cases: (1) the body has finite heat capacity and the ambient has infinite heat capacity; (2) both the body and the ambient have finite heat capacity. For the first case, the specific expression of Equation (12) is:(13)$$\int_{T_{b}(0)}^{T_{r}}C_{V}\left(\frac{1}{T_{r}}-\frac{1}{T_{b}}\right)dT_{b}+\int_{T_{r}}^{T_{b,\min}}C_{V}\left(\frac{1}{T_{r}}-\frac{1}{T_{b}}\right)dT_{b}=0,$$
where *C_V_* is the body’s heat capacity at a constant volume. Solving for Equation (13) leads to the same result as that given by Schilling et al.:(14)$$\left(\ln T_{b,\min}-\frac{T_{b,\min}}{T_{r}}\right)-\left(\ln T_{b}(0)-\frac{T_{b}(0)}{T_{r}}\right)=0.$$

Since the initial temperature of the body, *T*_b_(0), and the ambient temperature, *T*_r_, are given, Equation (14) is an implicit calculation formula related to the minimum temperature the body can reach, *T*_b,min_. According to Equation (14), when the initial temperature of the body is higher and the ambient temperature is lower, the minimum temperature that can be reached by the body is lower. Both a higher initial temperature of the body and a lower ambient temperature can create a larger positive transformation, which can compensate for the larger negative transformation corresponding to a larger reduction in the body temperature in the second stage.

Equation (12) can be further rewritten as:(15)$$\frac{1}{T_{r}}\left(Q_{1}+Q_{m}+Q_{2}\right)-\frac{1}{T_{b}}\left(Q_{1}+Q_{m}+Q_{2}\right)=0$$
where (Q_1_ + Q_m_ + Q_2_) is all the heat that moves away from the body and also all the heat that enters the ambient. All the heat that moves away from the body can be expressed as $-\int_{T_{b}(0)}^{T_{b,\min}}C_{V}dT_{b}$ and all the heat that enters the ambient can be expressed as $\int_{T_{r}(0)}^{T_{r,\max}}C_{V,r}dT_{r}$. Therefore, for the second case, the specific expression of Equation (12) is:(16)$$\int_{T_{b}(0)}^{T_{b,\min}}C_{V}\frac{1}{T_{b}}dT_{b}+\int_{T_{r}(0)}^{T_{r,\max}}C_{V,r}\frac{1}{T_{r}}dT_{r}=0.$$

By solving for Equation (16), we obtain:(17)$$C_{V}\ln\frac{T_{b,\min}}{T_{b}(0)}+C_{V,r}\ln\frac{T_{r,\max}}{T_{r}(0)}=0$$

Consider a special scenario in which the body has the same heat capacity as the ambient, *C_V_* = *C_V_*_,r_. Combining Equation (17) with the energy conservation equation easily yields:(18)$$\left\{\begin{array}{l} T_{b,\min}=T_{r}(0)\\ T_{r,\max}=T_{b}(0) \end{array}\right.$$

According to Equation (18), the minimum temperature that the body can reach is equal to the initial temperature of the ambient and the maximum temperature that the ambient can reach is equal to the initial temperature of the body. That is, the body swaps heat and temperature with the ambient. 

Schilling et al. claimed that their thermoelectric experiments bent the second law of thermodynamics. At the same time, they intended to show that their thermoelectric experiments did not violate the second law of thermodynamics by arguing for the entropy increase during the experiments. Schilling et al. did not clearly reveal the thermodynamic principles of their experiments, due to the misunderstandings about the second law of thermodynamics. First, the second law of thermodynamics as considered by Clausius is the theorem of the equivalence of transformations in reversible cycles, not the principle of entropy-increasing for irreversible isolated systems. The latter is based on the former. Second, irreversibility cannot be used to justify the supercooling phenomenon in Schilling et al.’s experiments. The supercooling phenomenon is achieved by a reversible combined cycle. The thermodynamic principles involved can be explained by using the theorem of the equivalence of transformations. Irreversibility is not a direct cause of the occurrence of supercooling, although irreversibility weakens it to a large extent. 

## 5. Conclusions

(1) In most present textbooks, the Clausius statement of the second law of thermodynamics is usually considered to be that “*Heat can never pass from a cold to a hot body without some other change*”. However, in fact Clausius viewed it only as a natural phenomenon. What Clausius originally referred to as the second law of thermodynamics was the theorem of the equivalence of two kinds of transformations, i.e., the transformation of work to heat and the transformation of heat from a high temperature to a low temperature in a reversible thermodynamic cycle.

(2) The theorem of the equivalence of transformations in thermoelectricity is established. There are two kinds of equivalent transformations in a reversible thermoelectric cycle, i.e., the transformation of heat from a high temperature to a low temperature and the transformation of electrical work to heat. As with a reversible heat–work cycle, the sum of the equivalence-values for all transformations in a reversible thermoelectric cycle is zero. According to the theorem of the equivalence of transformations, the same thermodynamic conclusions can be obtained in thermoelectric systems as in heat–work systems.

(3) The idea of a reversible cycle and the theorem of equivalence of transformations provide a thermodynamic basis for resolving thermoelectric problems. Kelvin’s thermoelectric circuit is essentially a reversible thermodynamic cycle. The well-known Kelvin relation can be derived from the theorem of the equivalence of transformations in a reversible thermoelectric cycle.

(4) The supercooling phenomenon in Schilling et al.’s experiments cannot be achieved by a single thermodynamic cycle, but needs to be achieved by a reversible combined power–refrigeration cycle. A reversible combined cycle can be fully expressed as a combination of several transformations of heat between different temperatures. According to the theorem of the equivalence of transformations, the occurrence of the negative transformation of heat at a low temperature to a high temperature must be compensated for by the positive transformation of heat from a high temperature to a low temperature. Not only does the supercooling phenomenon not bend the second law of thermodynamics, but it provides further compelling evidence of the second law of thermodynamics. Irreversibility cannot be used to justify the supercooling phenomenon. Irreversibility is not a direct cause of the occurrence of supercooling, although irreversibility weakens it to a large extent.

## Figures and Tables

**Figure 1 entropy-25-00155-f001:**
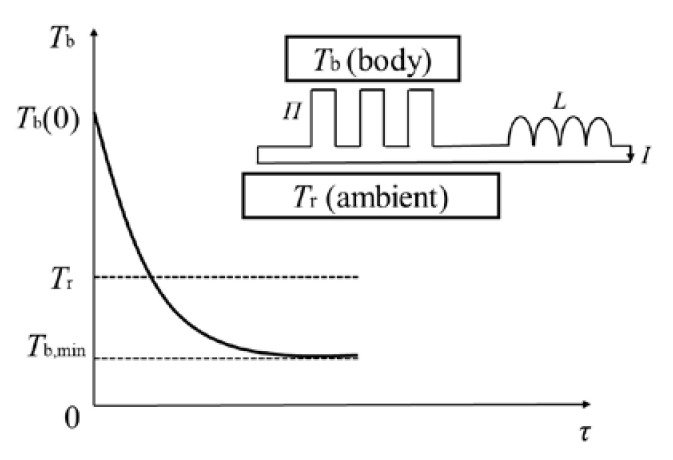
Schilling et al.’s thermoelectric experiments [1]. *T*_b_(0) denotes the initial temperature of the body, *T*_b,min_ denotes the minimum temperature the body can reach, *T*_r_ denotes the ambient temperature and *τ* denotes time. *Π* is the Peltier element, *L* is the electric inductance and *I* is the electric current.

**Figure 2 entropy-25-00155-f002:**
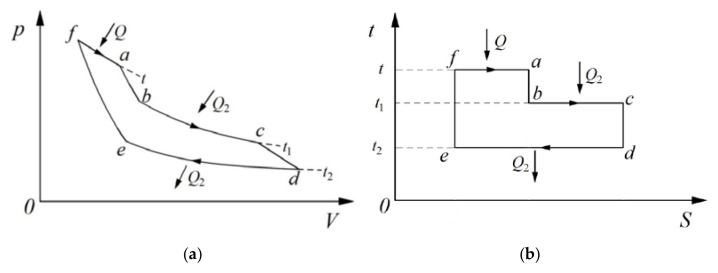
Thermodynamic cycle with three heat sources [5]. (**a**) *p-V* diagram; (**b**) *t-S* diagram.

**Figure 3 entropy-25-00155-f003:**
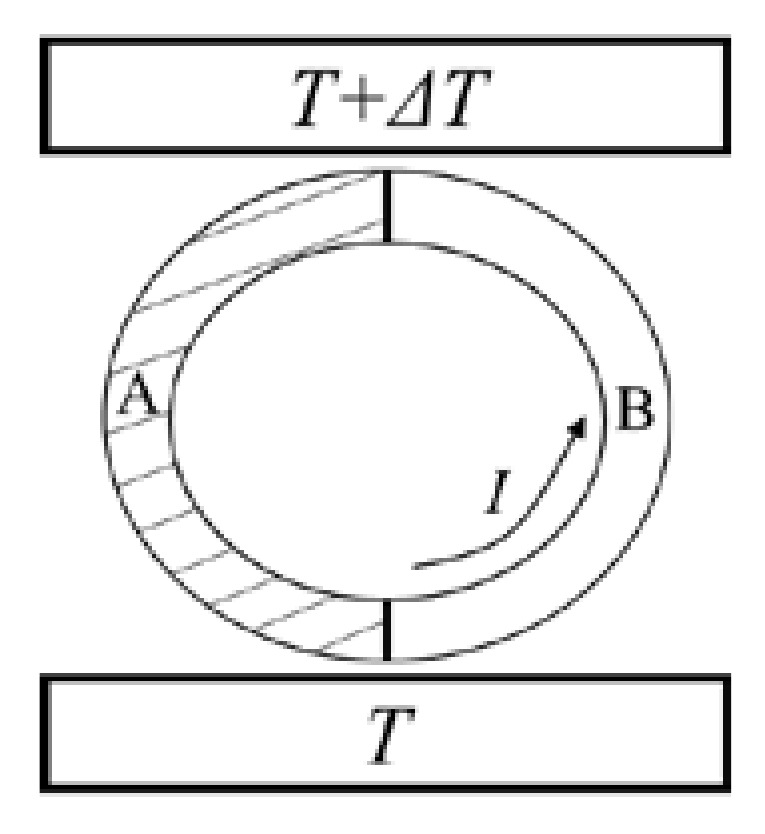
Kelvin’s thermoelectric circuit [21].

**Figure 4 entropy-25-00155-f004:**
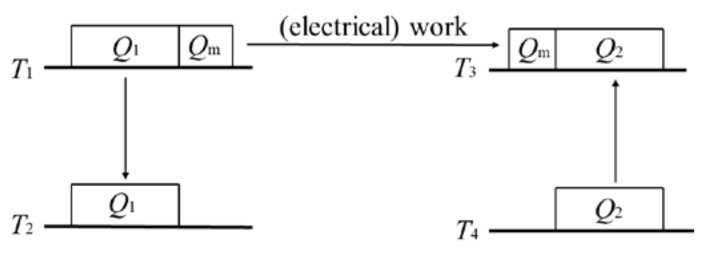
Schematic of transformations in a reversible combined power–refrigeration cycle.

**Figure 5 entropy-25-00155-f005:**
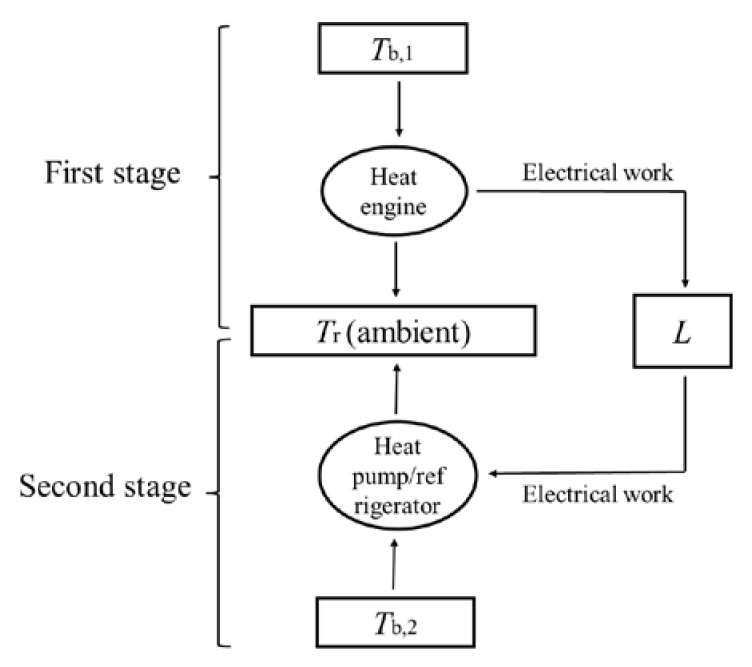
Combined cycle in Schilling et al.’s thermoelectric experiments.

**Figure 6 entropy-25-00155-f006:**
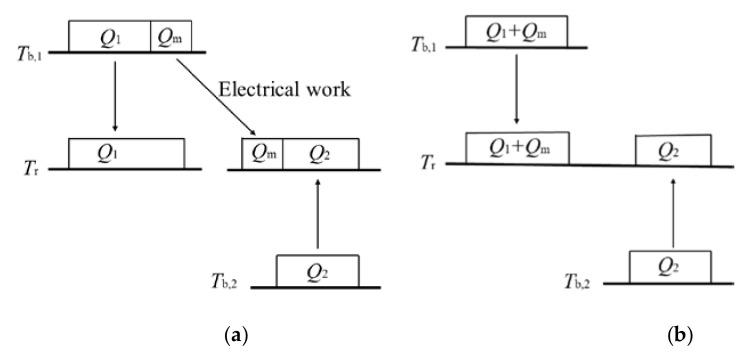
Schematic of transformations in Schilling et al.’s combined cycle. (**a**) Normal expression. (**b**) Simplified expression.

## Data Availability

Not applicable.

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
