# Peer review of "Thermoelectric Cycle and the Second Law of Thermodynamics"

_entropy, 2023, doi:10.3390/e25010155_

Round 1

Reviewer 1 Report

REVIEW (2022 December 23).
Thermoelectric cycle and the second law of thermodynamics by Ti-Wei Xue and Zeng-Yuan Guo*

The Manuscript proposes “the theorem of the equivalence of transformations for reversible thermoelectric cycles.” It also analyzes “the supercooling phenomenon by Schilling et al.[1R]” using  a reversible heat engine–heat pump/refrigerator combined cycle. They affirm that “the reduction of the body temperature to below the ambient temperature requires the body itself to have a higher initial temperature than the ambient as the ‘compensation’ necessary by the Second Law of Thermodynamics (SLT).”

The manuscript analysis is based on the original Clausius’ “Equivalence of Transformations (ET),” not widely used in the literature. In that regard, it is interesting and valuable to analyze from a different perspective the SLT’s elusive phenomena. However, the authors have probably overemphasized the ET without due rigor and at the expense of the more physical statements of the SLT. More specific explanations are needed to demonstrate the advantages of the analysis presented.

If the manuscript is streamlined and updated it will further contribute to the challenges of the SLT, and therefore justify publication in Entropy.

The followings are specific questions and suggestions for further clarification and improvement of the manuscript:

1.  Clausius himself, in his “Mechanical Theory of Heat,” titled CHAPTER IV as, “THE SECOND MAIN PRINCIPLE UNDER ANOTHER FORM, OR PRINCIPLE OF THE EQUIVALENCE OF TRANSFORMATIONS (ET).” The ET is not preferable but just “ANOTHER FORM” of the ‘Clausius struggle’ to fully comprehend the Carnot’s 1824 publication and deduce his “transformations” when heat is reversibly transferring from high temperature and in part releasing (converting to) work and in part transferring to low temperature. His reasoning was ingenious like Carnot’s, with debatable ET particulars, but with accurate final deductions of the “transformations’ equivalence-values,” with the f(t)=1/T integration factor, that resulted in the accurate definition of the entropy and definition of the quantitative relation of the SLT, namely the Clausius Equality for reversible cycles, and the Clausius Inequality as in the general form of the Second Law of Thermodynamics (SLT) that include entropy generation due to irreversibilities of different kinds.

2.       The authors have not stated that Clausius Eq.(1) is in-principle based on the Carnot’s “reversible equivalency” as presented in a more modern publication below [2R]. All ‘reasoning proofs’ regarding the definition of the absolute temperature (Kelvin, 1848) and entropy (Clausius, 1850s), as well as the Gibbs free energy (Gibbs 1878), were deduced from the ingenious and specific reasoning of maximum possible and equal efficiency of all reversible heat engine cycles, by Sadi Carnot in 1824, as explained in [2R, p 333-347]: “Carnot's deduced that all reversible processes between specific thermal reservoirs (t1 &t2) must be the most and equally efficient (reversible equivalency) or the ‘[Carnot's] caloric’ (but actually nonequilibrium work-potential) will be created (like heat from cold to hot without ‘compensation’) instead being conserved in reversible processes.

3.       The paper by Schilling et al. describes experimental results and limiting theoretical analysis. The manuscript is not distinguishing the two sets of results, although the experimental results were highly irreversible and inefficient, see comment 18. below. Appropriate comments should be clearly stated by the authors.

4.       In the manuscript the Thermoelectric Peltier (TE) cycle is referred sometimes unnecessarily as “thermocouple cycle.” Furthermore, the TE cycle could be as a TE Power cycle when heat is transferred from high to low temperature and work extracted, or as a TE Refrigeration cycle when work is utilized to transfer heat from low to high temperature. The latter distinction is not always fully clarified in the manuscript when general concepts are presented (heat-to-work or work-to-heat were sometimes confused), although it is properly stated when the combined cycle is described.

5.       Lines 86 & 93: “transformation of work to heat” is stated. Should it be ‘heat to work’ in the power cycle when “heat is transformed from high to low temperature”?

6.       Lines 132-133: “equivalence of transformations refers to qualitative…” However, the integration factor f(t)=1/T provides quantitative relation for the “transformation’s equivalence-value” and was used for entropy definition and quantification. Also, with entropy generation quantification (Sgen), the Clausius Inequality becomes specific equality: Cycle_Integral[dQ/T]=Sgen (>=0)

7.       Lines 138-140: “This statement [heat never pass from a cold to hot …] was only used by Clausius to prove the theorem of the equivalence of transformations [5].” Needs some clarification of how …”

8.       Lines 155-158: ”… two kinds of transformations … (a) heat from high to low temperature, (b) the transformation of electrical work to heat. These … phenomena of the same nature and are thus equivalent.” The (b) imply a refrigeration cycle, then (a) should be “heat from low to high temperature.”  Or vice versa? Not clear why “being of the same nature imply equivalency.” More appropriate justification is desired.

9.       Lines 165-169: “Since … equivalence transformations apply to both … This explains why thermoelectric system follows the same … as heat-work system.” Confusing explanation (since it applies…it explains why it applies)?

10.   Line 201: “…how Kelvin gets the correct Kelvin [or Clausius?] relations.” Should it be Clausius relations?

11.   Line 218: “thermal transformer” is cited without any reference. It was introduced in reference [3R, 2004] with a combined power-refrigeration cycle and relevant correlations. It was detailed in more detail in [4R] below, cited as [4] in the manuscript.

12.   Line 224 Eq.(10) & Fig.4: First, the TE power cycle is from T1 to T2 with Qm*F(T1,T2), and the refrigeration cycle from T4 to T3 with Qm*F(T4,T3). The authors should describe how the two heat-to-work transformations resulted in Qm*F(T1,T3) in Eq.(10) since the nomenclature is not reconciled. There is Nomenclature confusion. For example: On Fig.1: Tb(0), Tr, Tb,min. On Fig. 2: t, t1, t2. On Fig.4: T1, T2, T3, T4. Nomenclature from different sources should be clarified and reconciled. Parallel T-S diagrams where appropriate should be provided to better depict the cycle processes and undergoing physical phenomena.

13.   Lines 235, 241, etc.: “thermocouple circuit” should be “thermoelectric (TE) circuit.”

14.   Lines 248-268, Eqs(11&12), and Fig.6: “… combined cycle contains three transformations …” However, it consists of the two cycles (TE power cycle and TE refrigeration cycle) each with two transformations, thus the combined cycle should results in four transformations, with an equal magnitude of the two works. It appears that Qm*F(Tb2,Tr) is missing in Eq.(11) and elsewhere. Even if Qm is the same the relevant temperatures and thus the related transformations should not be the same. If a term is missing in Eq.(11), then Eq.(12) should be verified also.

15.   Line 253: “Heat source” is supplying heat into and “heat sink” is removing heat from a device. In general, they could be called “heat reservoirs” or “heat sinks” where appropriate.

16.   Line 284 and Eq.(15): How is Eq.(15) derived from Eq.(12)? Could case (1), Eq.(13) be derived as a special case of (2), Eq.13, when Cv=infinity and/or Tr=constant?

17.   Lines 300-303: The theorem of “equivalence of transformation (ET)” is not more superior than the “Principle of entropy conservation for reversible processes, nor than the entropy increase due to irreversibilities.” The ET was used to define and quantify entropy, and only after the entropy was defined, the quantitative generalization of the SLT was established, including the conservation of entropy in ideal reversible processes, entropy generation in real irreversible processes, and impossibility of entropy destruction by any process.

18.   Lines 303-307: Confusing statements contradict the Schilling’s experimental results. “The supercooling phenomenon in Schilling et al.’s experiments is achieved by a reversible combined cycle” is not accurate since the experimental results were highly inefficient and highly irreversible as stated by Schilling et al.[1], and as detailed and quantified by Kostic [4, Table 1], the same reference [4R] below: The experimental subcooling of only about 2C was achieved (compare with 69C subcooling if reversible processes), implying only 0.1% overall irreversible efficiency. “The thermodynamic principles involved can be clearly explained by using the theorem of the equivalence of transformations.” Not clear how specifically, need to justify.

19.   The Conclusions should be updated accordingly after the above is clarified and updated.

References used in this review:

[1R] Schilling, A.; Zhang, X.; Bossen, O. Heat flowing from cold to hot without external intervention by using a “thermal inductor”. Sci. adv. 2019, 5, eaat9953.

[2R] Kostic, M. Revisiting the Second Law of Energy Degradation and Entropy Generation: From Sadi Carnot’s Ingenious Reasoning to Holistic Generalization. AIP Conf. Proc. 2011, 1411, 327. [CrossRef]

[3R] M. Kostic, Irreversibility and Reversible Heat Transfer: The Quest and Nature of Energy and Entropy, IMECE2004, ASME Proceedings, ASME, New York, 2004. [CrossRef]

[4R] Kostic, M. M. “Heat Flowing from Cold to Hot without External Intervention” Demystified: Thermal-Transformer and Temperature Oscillator. arXiv preprint 2020, arXiv:2001.05991.

Reviewer 2 Report

The present manuscript is a criticism of the work of Schilling et al. It is an interesting study and can clarify the second law of thermodynamics more, for audiences. For this reason, I can recommend it. However, I want to assert that the Clausius statement mentioned in this manuscript is incomplete. The authors presented this statement: "heat can never pass from a cold to a hot body without some other change". However, the correct form of this statement (at least what has been stated in the famous textbooks) has the condition that it must be cyclic. One can use the stored energy to pass the heat to a colder temperature. But all know that after the stored energy gets finished, the passing heat to colder a temperature vanishes. The authors have pointed unconsciously to the cyclic condition in Eq. (4). As a result, before accepting the manuscript, the author should clarify why they havened deleted the condition of "being cyclic" from the Clausius statement.

Round 2

Reviewer 1 Report

The authors updated and improved the manuscript as per the reviewer's comments/questions and it is now suitable for publication. The authors should review the last version for any typos or other inconsistencies, and I do not need to see the final update.

This is an interesting analysis from another angle of the Second Law of Thermodynamics (SLT) for the Thermoelectric cycle and a critique of the Schilling et al. paper with an inappropriate dramatization of the "bending the Second Law."

It is hoped that the manuscript will contribute to the further clarification of the SLT and inspire further debate regarding the fundamental laws.